# Drug-Induced Autoimmune Hepatitis by Varenicline and Infliximab as a Continuous Disease Spectrum with Two Different Flares: Acute Liver Injury Followed by Hepatic Autoimmunity

**DOI:** 10.3390/ijms26199574

**Published:** 2025-09-30

**Authors:** Rolf Teschke

**Affiliations:** 1Department of Internal Medicine II, Division of Gastroenterology and Hepatology, Klinikum Hanau, D-63450 Hanau, Germany; rolf.teschke@gmx.de; 2Academic Teaching Hospital of the Medical Faculty, Goethe University Frankfurt/Main, D-60629 Frankfurt am Main, Germany

**Keywords:** DIAIH, idiosyncratic DILI, autoimmune hepatitis, updated RUCAM, simplified AIH score, varenicline, infliximab, CYP

## Abstract

Drug-induced autoimmune hepatitis (DIAIH) is a rare and complex disorder caused by drugs that are commonly metabolized by hepatic microsomal cytochrome P450 (CYP) pathways. Whereas DIAIH presents generally with a single clinical flare, in rare cases its clinical course shows two different, consecutively emerging flares. The aim of this report was to analyze details of this rare but interesting phenomenon and to help improve appropriate causality evaluation in patients with suspected iDILI or DIAIH to provide better insight into the pathomechanistic steps leading the diseases. A clinical course with two flares was found in a DIAIH patient treated with varenicline, a smoking cessation drug, and in another patient experiencing DIAIH following intravenous application of infliximab used to treat ankylosing spondylitis. In both patients, the first flare was determined as a typical liver injury with increased serum activities of alanine aminotransferase (ALT) and normal titers of serum autoimmune parameters, classified as an acute liver injury analogous to idiosyncratic DILI (iDILI), with verified causality using a modified version of RUCAM (Roussel Uclaf Causality Assessment Method). After an interval of around two months from the cessation of varenicline use, the second flare emerged, as evidenced by increased serum ALT values now associated with newly increased serum autoimmune titers of antinuclear antibodies (ANAs), classifying this flare as hepatic autoimmune injury with verified causality for varenicline using the simplified autoimmune hepatitis (AIH) score. A similar clinical DIAIH course of a continuous disease with two flares was described for the second patient, who received infliximab and experienced an interval of one month between the first and second flare. Interestingly to note, neither varenicline nor infliximab is degraded via a CYP pathway, and the metabolic disposition of both drugs is low. In sum, DIAIH can develop with two consecutive flares caused by two drugs not metabolized by CYPs and with slow drug disposition, raising the question of whether this phenomenon of two flares can occur in additional cases of DIAIH due to other drugs metabolized by CYPs or non-CYPs, a question to be resolved by DILI experts in future cases of iDILI and DIAIH.

## 1. Introduction

Immune idiosyncratic drug-induced liver injury is known as a group of four diseases lacking homogeneity and categorized as types 1–4 [1]. In this context, drug-induced autoimmune hepatitis (DIAIH) represents type 1, the most important type [2], followed by HLA-based drug-induced autoimmune liver injury, type 2, considering the genetics of HLAs (human leucocyte antigens) [3] based on several reports [4,5,6,7,8,9,10,11,12,13,14,15,16,17,18,19,20], idiosyncratic drug-induced anti-CYP autoimmune hepatitis, type 3, with a focus on hepatic microsomal cytochrome P450 (CYP) and its serum antibodies [21], as found in various single cases or case reports [22,23], and finally immune-based iDILI with a continuous disease spectrum ranging from Stevens-Johnson syndrome (SJS) to toxic epidermal necrolysis (TEN), type 4 [24], derived from recent case studies [25,26,27]. The nomenclature of each type described above is based on published cases that were all assessed using the Roussel Uclaf Causality Assessment Method (RUCAM) [28,29,30] with completion for type 1 using the simplified autoimmune hepatitis (AIH) score, which is the short form derived from the expanded term of the simplified criteria of the diagnosis of the AIH [31], and for type 4 using the algorithm of drug causality for epidermal necrolysis (ALDEN) [32]. DIAIH is the most frequent type and must be differentiated from types 2–4 [1,2].

Published cohorts comprising DIAIH cases were confounded occasionally by missing causality assessment using RUCAM, the simplified AIH score, or even worse, forgetting both diagnostic algorithms [2]. Other reports mentioned non-drugs such as herbal medicines as suspected causatives of DIAIH, an incorrect approach, as herbs cause herb-induced liver injury (HILI) and not DIAIH. In general, these methodological shortcomings impair the quality of the published results and conclusions. To refine the molecular and clinical features of DIAIH, it is recommended to also use data of twelve DIAIH cases of perfect quality because of their assessment by both RUCAM and the simplified AIH score.

The aim of this article was to provide further insight into the molecular and mechanistic sequelae that occur in patients with verified DIAIH. A challenge is the fact that DIAIH consists of two components, each triggering the disease through different processes. In addition, the role of two consecutive events within the cascade leading finally to DIAIH is fascinating. The first step can be traced back to the direct injurious effects of the suspected drug or its metabolites, which disturb primarily intracellular homeostasis and in addition generate molecular macro-complexes as haptens with antigen properties. In the second step, antigenicity allows for the activation of the innate immune system to the adaptive immune system responsible for the clinical course, exemplified as autoimmunity. The treatment efficacy of immunosuppressive agents such as corticosteroids is due to amelioration of the final autoimmune processes.

## 2. Strategy of Literature Search

The PubMed database and Google Science were used for the literature search. The following terms were used: cytochrome P450, haptens, DIAIH, autoimmune DILI, immune-mediated DILI, DILI with autoimmune features, DIALH, and AIH. The search was completed on 12 August 2025. Papers were preferred that used validated causality assessment algorithms.

## 3. Causality Assessment of Drug-Induced Autoimmune Hepatitis

To ascertain the causality of any drug suspected to be responsible for initiation of DIAIH, the use of RUCAM and the simplified AIH score as robust causality assessment methods is mandatory. Each algorithm covers one of the two disease segments, supported by bimodal diagnostic criteria that help provide a final score for overall causality grading [28,29,30,31]. Neglecting this approach is outside of good clinical practice and leads to debated results that do not warrant further discussion [2].

### 3.1. RUCAM

RUCAM in its original version [28,29], or better, its update of 2016 [30], is the preferred method for evaluating causality for suspected drugs in the frame of the initial injurious part of DIAIH. The advantages of RUCAM include internal method validation [29] supported by subsequent external validation [23,33,34], as summarized [35], while missing validation of other tools was criticized [30,35]. RUCAM was acknowledged by the US LiverTox database, which classified the RUCAM system as a method of assigning points for clinical, serological, biochemical, and radiological characteristics of liver injury [36]. The database notes that the RUCAM system provides an overall assessment score by reflecting the likelihood that the hepatic injury occurred due to a specific medication [36]. In addition, it confirmed that RUCAM is now widely used to assess the causality of DILI, both in the published literature and in support of regulatory decisions regarding medications implicated in causing hepatic injury. The LiverTox database also specified that RUCAM has been evaluated for accuracy, reproducibility, and intraobserver variability. Because the RUCAM score is based upon objective criteria, there should actually be little or no variation in the final scores obtained by different investigators, as clarified by the LiverTox database [36]. Indeed, RUCAM is also known for its transparency [21,28,29,30,35], liver injury specificity [28], and objectivity [30,35]. RUCAM with its scoring system provides causality gradings from excluded to highly probable [30]. A few limitations have been described if assessors neglect the principles of good clinical practice, as evidenced by issues of documented case data manipulation [35].

### 3.2. Simplified AIH Score

The simplified AIH score is again a validated diagnostic algorithm, but directed to the autoimmune specifics of DIAIH [31]. It is a scoring system used worldwide that provides causality gradings from excluded up to definite. For reasons of completeness and in the context of assessing causality in patients with suspected AIH, additional methods were proposed, the rarely used update of the simplified criteria for autoimmune hepatitis [37] and the International Autoimmune Hepatitis Group (IAIHG) score [38]. Nevertheless, DIAIH cases are still best assessed by the simplified AIH score [31] after use of the updated RUCAM [30], with causality gradings of at least probable evaluated by both diagnostic algorithms [2].

### 3.3. Sequential Use of the Diagnostic Algorithms

In view of the observation that not all suspected drugs fulfill the criteria of DIAIH, a cautious stepwise use of causality assessment methods is recommended. This procedure ensures that the liver biopsy, which is an invasive diagnostic procedure in the setting of the simplified AIH score, comes at the very end of the diagnostic evaluation, when the final decision has to be made on the validity of the DIAIH diagnosis. Therefore, the updated RUCAM [30] should be used at first to evaluate the drug-induced acute liver injury (ALI) part of DIAIH [2]. Only if increased titers of serum anti-nuclear antibodies (ANAs) and anti-smooth antibodies (ASMAs) associated with high serum immunoglobulin G (IgG) levels are found, should the case assessment proceed to and be finalized by the simplified AIH score [31]. This stepwise approach helps reduce expenses and risks for the patient by minimizing laboratory costs and liver puncture. At variance with other methods, missing case data are considered, though with score deductions, by both the updated RUCAM [30] and the simplified AIH score [31], a significant advantage making arbitrary opinion unnecessary. Missing data in AIH parts of DIAIH can refer to seronegativity of serum autoimmune parameters and IgG levels.

### 3.4. Approaches Unifying Diagnostic Causality Assessment Algorithms

In an attempt to facilitate causality assessment in DIAIH cases, it was proposed that the two causality algorithms be replaced by a single one through integrating the updated RUCAM in the IAIHG scoring system, but the practicability of this approach was not evaluated in real DIAIH cases [39]. Another attempt to obtain a single algorithm through combining parts of a modified RUCAM with parts of a modified IAIGH score was not further evaluated because of missing DIAIH cases due to specific drugs [40]. Despite these seemingly innovative theoretical considerations, the updated RUCAM and the simplified AIH score remain cornerstones in the evaluation of suspected DIAIH cases.

## 4. Definitions

### 4.1. Drug-Induced Autoimmune Hepatitis (DIAIH)

DIAIH is best described by its two features: one is characterized by the acute liver injury effects of the drug or its metabolites, while the second one reflects liver injury due to autoimmune reactions with elevation of serum autoimmune titers [2]. There is a clinical need for any case of suspected DIAIH to be differentiated from iDILI cases for which apparent serum autoimmune parameters are missing [30]. A clear differentiation is also mandatory for patients with idiopathic AIH that is characterized by increased titers of serum autoimmune parameters, but is not caused by drugs, while a variety of causatives are under discussion [31,37,38].

### 4.2. Idiosyncratic Drug-Induced Liver Injury (iDILI)

RUCAM in its original and updated version [28,29,30] helped define characteristics of iDILI in 81,856 cases published until mid-2020 [41]. The updated RUCAM was applied and specifically mentioned in the title of several more recent iDILI studies [42,43,44,45,46,47,48,49,50,51,52,53,54,55,56,57,58,59,60,61,62,63,64,65,66,67,68,69,70,71,72,73,74]. Due to its global use and appreciation, RUCAM became the reference standard in suspected iDILI cases to be assessed for causality [28,29,30,31,32,33,34,35,36,37,38,39,40,41,42,43,44,45,46,47,48,49,50,51,52,53,54,55,56,57,58,59,60,61,62,63,64,65,66,67,68,69,70,71,72,73,74]. As a rare disease, iDILI develops in a few genetically predisposed individuals following treatment with conventional drugs used in normal daily doses [75]. It is essential to differentiate unpredictable iDILI from intrinsic DILI that may be caused in any person independently of genetics after use of high amounts of a drug, with paracetamol the most frequently described medicine [75,76,77]: acetaminophen (paracetamol) overdose is the most cited example of intrinsic DILI [75,76,77,78,79].

### 4.3. Idiopathic Autoimmune Hepatitis (AIH)

Laboratory data on AIH include various serum autoimmune parameters with increased titers, especially ANA and ASMA syn smooth muscle actin (SMA) [2,31] or syn anti-smooth muscle antibody (SMA) [80], which are in addition to increased serum immunoglobulin G (IgG) levels among the key diagnostic elements of the simplified AIH score [31]. Causes of AIH are idiopathic, syn indeterminate, or undetermined, with clear exclusion of drugs as triggering compounds [31,80], except in cases of DIAIH [2,81]. In this context, concern has been expressed that AIH can hardly be differentiated from DILI [80], an argument difficult to reconcile when the updated RUCAM [30] and the simplified AIH score [31] are applied [2]. However, it has convincingly been outlined that AIH develops preferentially in patients of advanced age, requiring treatment by drugs possibly long before AIH was diagnosed and implicating drugs as a possible cause of AIH [80]. It is under discussion whether possible causatives are genetic risk factors and infections by various viruses, such as parvovirus B19 in children and hepatitis E virus in adults [80,81]. Clinical manifestations range from subclinical mild intermittent activity elevations of ALT or ALP with or without jaundice to cirrhosis, fulminant hepatitis, and acute liver failure [80].

## 5. Quality of DIAIH Cases Essential for Analysis

### 5.1. Search for Appropriate DIAIH Study Cohorts

There have been DIAIH cases of variable quality published [2]. Although known as a separate disease different from other common liver disorders for more than a decade, a uniform approach to how best to verify causality in suspected DIAIH cases is lacking. To facilitate descriptions of DIAIH characteristics, cases assessed by both RUCAM and simplified AIH score are essential. In line of these considerations, an analysis revealed that DIAIH cases published in 12 reports fulfilled the inclusion criteria of dual-causality assessments (Table 1) [82,83,84,85,86,87,88,89,90,91,92,93].

Serum autoimmune parameters are part of the simplified AIH score [31] and assist in supporting the diagnosis of DIAIH [38]. Among these parameters are IgG, ANA, ASMA, and SLA, but often they are not specifically mentioned in reports in reference to the drug suspected of being involved in DIAIH. ANA is the most frequent autoimmune parameter found, with positive titers in 77.3% of DIAIH patients [54]. Laboratory data and autoimmune parameters reported for a few patients with DIAIH caused by selected drugs are listed in Table 2 [2].

Data of treatment options and outcomes of DIAIH cases were published in five reports for specific drugs [82,83,88,89,93], with verified DIAIH diagnoses using RUCAM [28,30] and the simplified AIH score [32]. Accordingly, most patients received immunosuppressive treatment with a preference for prednisolone, leading to complete remission of clinical signs and laboratory results (Table 3).

A remarkable observation is that a few DIAIH patients experienced complete remission just by cessation of the causative drug (Table 3). Remission by drug cessation was only found in cases of DIAIH due to adalimumab [82], atorvastatin [89], infliximab [83], and sorafenib [88]. The reason for this kind of remission remains unclear, but may be due to a low-severity grade of the hepatic autoimmune injury, possibly associated with sufficient amounts of protective antioxidants in the liver.

### 5.2. Cautionary Note on Unqualified DIAIH Cases Due to Incomplete Causality Assessment

A previous analysis revealed that among 20 DIAIH publications, 12/20 reports (60%) were identified as being perfectly evaluated by both RUCAM and the simplified AIH score to verify causality in the reported DIAIH cases, confirming thereby the initially suspected diagnosis as correct [2], now listed (Table 1). On the contrary, in 4/20 reports (20%), only RUCAM was used, 2/20 reports (10%) applied only the simplified AIH score, and 2/20 reports (10%) presented cases lacking any causality assessment [2]. Thus, a cautionary note is warranted to identify cases of suspected DIAIH that remained causally unassessed by appropriate methods (Table 4).

The use of RUCAM was perfect in suspected DIAIH cases by ten members of the Drug-Induced Liver Injury Network (DILIN), who now used the validated original RUCAM [94] over the unvalidated US DILIN method, the tool conflicted by many shortcomings of which have been listed previously [35]. However, all ten DILIN authors forgot to assess the AIH part of their DIAIH cases as a result of neglecting to use the simplified AIH score, first available in 2008 [31] and thereby five years before the DILIN report was published in 2013 [94]. Although the RUCAM approach by DILIN members was only partially perfect [94], it paved the way for an increasing tendency among US scientists to use RUCAM in many other DILI reports and review articles [4,11,30,35,102,103,104,105,106,107,108,109,110,111,112,113,114,115,116,117,118,119,120,121,122,123,124,125,126,127,128,129,130,131,132,133,134,135]. Some of the US reports were included in the list of the worldwide 81,856 DILI cases all assessed by RUCAM as published until mid-2020 [41], and are mentioned above (Table 1).

## 6. Continuous Disease Spectrum of DIAIH with Two Injurious Flares

### 6.1. Basic Aspects

“Continuous disease spectrum” is a rare term used in clinical medicine. As an example, it describes the steady progress of the mild drug-dependent Stevens-Johnson syndrome (SJS) to the severe, newly named toxic eruptive necrolysis (TEN) [1,24]. On the contrary, DIAIH is virtually unique in the sense of continuous diseases due to its common restriction to the first contact of DIAIH patients with a physician in general practice or in the hospital, resulting in important case details during the subsequent clinical course mostly unpublished.

### 6.2. DIAIH Due to the Smoking Cessation Agent Varenicline

Based on an excellent clinical and diagnostic analysis, a report from Japan was published focusing on the two-step progression of varenicline-induced autoimmune hepatitis [136]. This breakthrough publication was largely neglected by the DILI community, including our group, likely because it was not readily available with open access. In this report, a female patient was diagnosed with drug-induced hepatitis due to the smoking cessation agent varenicline. This was described in detail, and the liver injury progressed in two steps [136]. As this case report represented a new and promotional reference approach to better follow initial cases of iDILI progressing eventually to DIAIH, several points and many essential details warrant further mention, as briefly summarized derived from the report [136]: (1) the liver injury started 5 days after daily treatment with varenicline in recommended doses with increased serum ALT values of 886 U/L and ALP of 419 U/L, as well as normal total bilirubin values of 1.3 mg/dL, in the absence of viral/autoimmune responses, whereas withdrawal of varenicline and treatment with ursodeoxycholic acid lowered the increase in the levels of liver enzymes immediately, as shown in their Figure 1, with ALT of around 80 U/L and ALP of around 200 U/L; and (2) surprisingly, the patient was readmitted to the hospital four weeks after the previous hospitalization because of increased aminotransferases detected in the course of a control examination, and while physical evaluation was again unremarkable, ALT was 588 U/L and total bilirubin 0.7 mg/dL, but serum ANA titers became now positive and signified new AIH features under conditions of unchanged serum IgG levels.

Causality assessment for the first flare [136] was achieved by the Digestive Disease Week-Japan (DDW-J) scale for DILI [137], a modified RUCAM tool commonly used in Japan, which assigned a score of 7 for probable causality for varenicline and classified the first flare as a typical iDILI [136]. Causality assessment of the second flare was evaluated using the simplified AIH score [31], yielding a score of 6 for probable causality, whereas this score was 2 at the first flare and thereby non-diagnostic [136]. In addition, histology provided strong support for the new development of AIH features, as evidenced by interface hepatitis with lymphocytic and lymphoplasmacytic portal inflammatory infiltrates extending into lobules. As expected, treatment with corticosteroids in tapering doses was effective for the second flare of the disease triggered by autoimmunity. The clinical course, laboratory results, and liver histology specifics confirmed DIAIH as the final diagnosis and provided new information in that varenicline initiated the first liver injury, resulting in the first flare of the DIAIH, followed by the second flare, attributed to evolving autoimmunity [136]. Notably, scientific advisors of the US LiverTox database, with NIH as financier, included the varenicline case as a smallipping narrative without own analysis of the exciting clinical news, mentioning merely a few case details, such as elevations in serum enzymes while on varenicline, which improved upon stopping, but rose again 8 weeks later with the appearance of ANA positivity, ultimately requiring corticosteroid therapy. However, the database report ignored data on causality assessments and failed to classify the case as DIAIH, likely due to overlooking or ignoring new developments coming up in this specific field [138]. Such analytical shortcomings of the LiverTox database were not new, as also seen in the context of other methodological pitfalls in reference to clinical evaluations and causality assessments of cases of iDILI included in the database, bringing into question its promoted pivotal role in the DILI field [139].

For reasons of completeness, all other published cases of iDILI due to varenicline were limited to an initial flare of acute liver injury without any subsequent autoimmune flare, thereby excluding DIAIH as final diagnosis [108,140,141]. Another example is a male patient who experienced new-onset jaundice, nausea, and fatigue associated with increased serum ALT of 1592 U/L, ALP of 254 U/L, total bilirubin of 12.0 mg/dL, negative ANA and ASMA, and normal immunoglobulin G within 5 days of starting a treatment with varenicline. Varenicline exposure was consequently stopped [108]. Around two months after cessation of the varenicline use, serum ALT and ALP had normalized with near normalization of total bilirubin, seemingly without a second flare. A probable causality grading for varenicline was achieved using RUCAM.

In another case report, a female patient suffered from jaundice, pruritus, dark urine, nausea, and fatigue after three weeks on varenicline, and one week after she stopped taking the medication, her serum ALT was 466 U/L, ALP 249 U/L, total bilirubin 5.61 mg/dL, and negative ANA and ASMA, with a return to normal values within 1–2 weeks [140]. A second flare was not reported, classifying this case as typical iDILI. A RUCAM score of 6 was obtained, in line with a probable causality grading for varenicline [140]. This RUCAM-based case was mentioned in the LiverTox database and qualified as reasonably convincing [138].

In a third male patient with underlying alcoholic liver disease and a history of hepatitis C virus infection, elevated serum ALT and ALP values were recorded, with the liver injury manifesting four weeks after initiation of varenicline use [141]. Cessation of varenicline use led to a normalization of ALT within four months and of ALP within one month. Applying the RUCAM, causality for varenicline was deemed probable.

Analysis of all cases revealed that varenicline caused iDILI in a single flare, verified by RUCAM [108,140,141], while two flares were observed in only a single case [136].

### 6.3. DIAIH Due to TNF-α Antagonist Infliximab

Two flares were described in a male patient with quiescent Crohn’s disease due to use of intravenous infliximab who had ALT of 1270 U/L at the first flare [94]. After drug cessation, ALT fell to 198 U/L by two months after the last infusion, but rose again to 1167 U/L. Initially, the serum ANA titer was negative, but became positive one month later at the second flare. Prednisone was started, and serum ALT normalized within 2 months. This was associated with his serum ANA, which reverted to negative. Using RUCAM, possible causality for infliximab was calculated, while attempts to apply their DILIN methods remained highly questionable, as this tool lacks internal and external validation [35]. Difficult to reconcile was the fact that the simplified AIH score was not used; therefore, the case was categorized as only possible DIAIH. At the time of publication of this infliximab report in 2013, the eleven authoring DILIN members obviously were not yet familiar with DIAIH and the requirements to obtain a robust diagnosis based on evidence such as the simplified AIH score.

## 7. Molecular and Mechanistic Sequelae of the Two Flares Caused by Varenicline in the DIAIH Cases

It is well documented in the literature that many drugs are capable of causing iDILI [30,41], and some of these drugs can trigger also DIAIH (Table 1) [2]. This condition applies, for instance, to infliximab, which is known to cause iDILI [41,94] as well as DIAIH [83,84,85,91,92,94]. This is applicable also to varenicline, because its use can lead to iDILI [140,141,142] or to DIAIH [136]. The observation of two flares observed during the development of DIAIH is a common feature of both drugs: infliximab [94] and varenicline [136].

### 7.1. Acute Liver Injury by Varenicline as First Flare of DIAIH

#### 7.1.1. General Metabolic Aspects of Varenicline

Varenicline is a selective partial agonist of the α(4)β(2) nicotinic acetylcholine receptor, chemically C_13_H_13_N_3_ with a molecular weight of 211.26 and a drug approved by the US FDA to assist smoking cessation [142,143]. With an organic heterotetracyclic structure and as a bridged as well as secondary amino compound, the chemical structure of varenicline is complex, consisting of several rings with the inclusion of nitrogen (N) and five unsaturated bonds (Figure 1).

After oral varenicline intake, absorption is virtually complete and systemic availability is high. Varenicline is almost exclusively excreted unchanged in urine, primarily through glomerular filtration, with some component of active tubular secretion via the human organic cation transporter (hOCT-2) [142,143]. The usual oral dosage in adults is 1 mg twice daily for 12 weeks, with an initial titration week. Varenicline does not undergo significant metabolism, and is not metabolized by hepatic microsomal CYP enzymes. Consistent with an elimination half-life of around 24 h, steady-state conditions are reached within 4 days of repeat dosing of the α_4_β_2_ nicotinic acetylcholine receptor. Varenicline is a drug approved by the US FDA to assist smoking cessation [142]. Consensus exists that varenicline is not a substrate of CYP or metabolized by CYP. The pathogenesis of the DIAIH remains to be elucidated [136,138,142,143]. As a result, varenicline falls in the group of drugs that can initiate iDILI and is metabolized by non-CYP pathways [144].

#### 7.1.2. Non-CYP Pathways in iDILI

It is remarkable that liver injury by varenicline [108,136,138,140,141] occurs despite missing metabolism via one of several cytochrome CYP isoforms [136,138,142,143]. This is in contrast to RUCAM-based iDILI due to at least 28/48 drugs (58.3%), for which clinical or experimental evidence exists that the metabolism proceeds via CYP, whereas for other iDILI cases due to the remaining 20 drugs (41.7%), there were negative or missing results implicating CYP in the metabolism of these drugs [144]. Accordingly, there is a group of drugs that can injure the liver in the absence of metabolism via CYP. Among these drugs are, in addition to varenicline [136], allopurinol [145], amoxicillin-clavulanate [146], azathioprine/6-mercaptopurine [147], busulfan [148], dantrolene [149], didanosine [150], floxuridine [151], hydralazine [152], infliximab [138,153], interferon alpha/peginterferon [154], interferon beta [155], ketoconazole [156], methotrexate [157], minocycline [158], sodium aurothiomalate [33], nitrofurantoin [159], pyrazinamide [160], rifampicin [161], sulfasalazine [162], and thioguanine [163].

#### 7.1.3. Role of Hepatic Non-CYP Enzymes in Acute Liver Injury by Varenicline

As hepatic microsomal CYPs are not available for metabolism in a group of at least 20 drugs (41.7%) with the potential of causing iDILI confirmed as robust diagnosis through verification by RUCAM, hepatic non-CYPs must take over as important pathways metabolizing these drugs [144,164,165,166,167,168,169], including now varenicline, which is not metabolized by CYPs [136,138,142,143]. Not limited to drugs causing iDILI [144], but in a broader context, approximately 30% of clinically used drugs are metabolized by non-CYP enzymes [169]. Varenicline tartrate (7,8,9,10-tetrahydro-6,10-methano-6*H*-pyrazino [2,3-h][3]benzazepine) is largely excreted as unchanged drug and was identified together with its metabolites in the urine of patients exposed to varenicline [142,164]. Among the detectable metabolites were those that arose via oxidation and N-carbamoyl glucuronidation, in addition to metabolites that were generated via N-formylation and formation of a novel hexose conjugate [164]. More specifically, these were oxovarenicline, N-glucosyl varenicline, N-glucosyl varenicline, N-formylvarenicline, 2-hydroxyvarenicline, and varenicline N-carbamoyl glucuronide catalyzed by UDP-glucuronosyltransferase-2B7 (UGT2B7) in human liver microsomes [142]. Based on these metabolic details [142,164], varenicline obviously undergoes phase 1 and phase 2 metabolic pathways in a sequential and interconnected manner, basically in a similar way to drug metabolism via CYPs [1,21,35,144,169]. While phase 1 is preferentially involved in oxidation processes, the phase 2 reaction is responsible for conjugation processes to make the drug excretable via urine or bile.

The non-CYP enzyme(s) responsible for liver injury by varenicline have not yet been completely determined, except for UDP-glucuronosyltransferase, which is needed for glucuronidation of the metabolites. Various non-CYP enzymes are known in the liver [144,165,166,167,168,169]. Among these are flavin-containing monooxygenase 3 and 5, alcohol dehydrogenase 1A, 1C, and 4, aldehyde oxidase 1, aldehyde dehydrogenase 1A1, xanthine oxidase, xanthine dehydrogenase, and monoamine oxidase A, well compiled in a recent report [169], in addition to ADP-ribosyltransferase, y-glutamyl transferase, cathepsin B [167], and carboxylesterase [168].

Considering various non-CYP enzymes, good candidates for metabolic varenicline disposition are human xanthine dehydrogenase (XDH) and xanthine oxidase (XO). The catalytic function of both enzymes is integrated in xanthine oxidoreductase (XOR), which also generates reactive oxygen species (ROS), a chemical by-product known to trigger liver injury in CYP-dependent reactions, during drug metabolism processes [144,170,171]. The main XOR functions are [170,171]: (1) XDH activity that promotes the last two steps of purine catabolism, from hypoxanthine to uric acid; (2) XO activity that, besides purine catabolism, generates ROS; (3) nitrite reductase activity that produces nitric oxide; and (4) NADH oxidase activity that generates ROS. All these XOR activities catalyze both phase 1 and phase 2 metabolic pathways and help metabolize endogenous and exogenous compounds, possibly including varenicline, among other drugs [171].

#### 7.1.4. Molecular and Mechanistic Sequelae Causing the First Flare of DIAIH by Varenicline

At the molecular level, varenicline must be metabolized by non-CYP enzymes due to the absence of functioning CYP. The preferred non-CYP enzyme is XOR, whereby varenicline is likely oxidized along with the generation of ROS and responsible for various effects, as follows. (1) In normal individuals without varenicline use, the reactive intermediates generated are commonly scavenged by hepatic antioxidants such as glutathione if hepatic antioxidant levels are not exhausted [172]. (2) In patients under varenicline therapy, some ROS will be needed for the phase 1 metabolic pathway to facilitate varenicline degradation, while the remaining ROS can be scavenged well, which is associated with prevention of liver injury. These conditions apply to the great number of varenicline users who in fact tolerate varenicline well and do not experience liver injury or slightly increased LTs, as noted by the LiverTox database, which specifies that varenicline has not been associated with rates of serum enzyme elevations during therapy greater than that occurring with placebo therapy, but information on these abnormalities is limited and occasional instances of asymptomatic ALT elevations leading to drug discontinuation have been reported [138]. (3) Among the three patients with verified iDILI related to varenicline use, excess ROS in the liver will cause iDILI by varenicline, presenting as a single flare of a typical iDILI excluding a DIAIH constellation due to missing overt signs of autoimmunity [108,140,141]. Finally, (4) in the single patient with confirmed DIAIH and two flares [136], ROS facilitated varenicline metabolism, but the excess ROS injured the hepatocytes, leading to the first flare of the DIAIH. At this early stage of the liver injury, the combined action of varenicline, its metabolites, and the injurious effects of the reactive intermediates can function in a cross-talk manner for initial steps directed to autoimmunity. This is not yet evidenced by increased titers of serum autoimmune parameters or enhanced serum IgG levels.

The first flare of DIAIH due to varenicline and ROS in the sense of a phase 1 metabolic pathway may theoretically be aggravated by phase 2 reactions due to UDP-glucuronosyltransferases (UGTs), N-acetyltransferases, or sulfotransferases through the formation of toxic reactive metabolites, as shown in a study comprising 317 drugs suspected to cause iDILI, but not verified by RUCAM [173]. In this context, aggravation may also be triggered by aryl glucuronides formed from carboxylic acid-containing drugs that have been proposed to be responsible for iDILI caused by carboxylic acids such as diclofenac [174]; however, the evidence is far from compelling [144] and remains a controversial issue among experts [175,176,177,178]. Whether carboxylic acids as aryl glucuronides are generated during varenicline degradation has not yet been studied.

It seems that ROS derived from non-CYP processes likely initiates the first flare of DIAIH observed following treatment by varenicline [136] through attacks of reactive intermediates on membrane structures of subcellular organelles within the hepatocytes in a similar way to that known for drugs that cause iDILI following metabolism by the CYP system [2,144]. During this first liver injury flare in the course of varenicline use, autoimmune events due to primarily triggering agents are unlikely or at least they are not accompanied by emerging autoimmune parameters, which were not found to have abnormal titers in the serum of the involved patient [136]. Another issue worthy of discussion is the context of the first flare [136]. The opinion has been expressed that in most iDILI cases, the hepatic immune system may play a prominent pathogenic role mediated by CD8 T cells of the adaptive immune system that requires prior activation of the innate immune system through danger-associated molecular pattern molecules (DAMPs), whereas a role of non-immune mechanism cannot be excluded [179,180]. For the first flare in the current varenicline case, the role of immune or autoimmune events was evidently not apparent. Instead, the autoimmunity process developed long after the first flare in the varenicline patient under consideration [136].

### 7.2. Autoimmune Liver Injury by Varenicline as Second Flare of DIAIH

#### 7.2.1. Tentative Compounds Triggering the Autoimmunity

Autoimmunity can be triggered by various compounds acting as antigens [80], traced back specifically to varenicline and its tentative metabolites in the patient with DIAIH due to varenicline use [136]. However, the individual neoantigen(s) responsible for the second flare of DIAIH by varenicline remained undetermined.

#### 7.2.2. Molecular and Mechanistic Autoimmune Processes

Autoimmunity was visible only at the second DIAIH flare, long after the use of varenicline for 5 days and long after the first flare of the disease [136]. Molecular and mechanistic pathophysiological processes are based on several hypotheses [80,88,181,182]: (1) the long interval between the first and second flare is attributable to the high systemic availability of varenicline due to its slow degradation [141,142]; (2) neoantigens are gradually formed in the hepatocytes through covalent binding of the modified parent drug or reactive varenicline metabolites and ROS with cellular proteins; (3) covalent binding occurs at the site of membrane constituents of liver mitochondria and the endoplasmic reticulum, which corresponds to the biochemical microsomal fraction; (4) neoantigens are responsible for the antibody response of release of autoantibodies from the liver to the systemic circulation; and finally, (5) as a result, immune reactions eventually manifest in a second flare of DIAIH with detectable newly emerging autoimmune parameters in the blood, as in the patient following varenicline treatment [136].

## 8. Conceptual Considerations Regarding the Two Sequential Flares of DIAIH by Varenicline and Infliximab

### 8.1. Varenicline

The Japanese case of two flares observed in a patient with DIAIH following varenicline use is a report of excellence [136]. This publication provides new information at least for varenicline that this drug is capable of causing DIAIH alongside two flares occurring separately from each other, a unique situation because current concepts favor the simultaneous occurrence of acute liver injury and the autoimmune reaction in cases of DIAIH with a diagnosis verified by RUCAM [30] and the simplified AIH score [31], as caused by virtually all listed drugs (Table 1) [82,83,84,85,86,87,88,89,90,91,92,93]. The long interval between the first flare and the second one implies a longer time to generate the autoimmune features to complete the injurious disruption as DIAIH [136]. With respect to other drugs, such long intervals may have been missed in cases of iDILI if after the initial diagnosis no clinical or laboratory follow-up was implemented to evaluate whether a new flare and now with increased titers of serum autoimmune parameters may have developed [136]. In this context, concern has been expressed that in patients with established genuine AIH, the priming of the immune system could occur years before the development of the AIH [80,183], converting the idiopathic or idiosyncratic genuine AIH to an explainable AIH [183].

In general, attributing an AIH case to a previous iDILI or DIAIH can be difficult because of the long interval and the reduced capacity to remember. Previous iDILI events may have been undetected if asymptomatic or in remission due to drug cessation alone, for instance, in cases of DIAIH (Table 3). In this context, the Japanese varenicline patient was primarily asymptomatic, and the increased ALT values were detected only by chance [136]. Similarly, a possible autoimmune flare can remain undetected due to undulation in serum liver tests or missing symptoms: in the Japanese patient under consideration, increased ALT values were found again long after the first liver injury and only by chance, which then allowed for description of the second flare.

The Japanese varenicline case provides another specific important detail, as serum ALT activities of the first flare were substantially higher compared with those of the second flare, which exhibited values 66% those of the first [136]. For the second flare, treatment by prednisolone was effective. Transferring this result to all DIAIH cases (Table 3) explains the efficacy of immunosuppressive agents in virtually all DIAIH cases through their positive effects, likely attributable to autoimmunity. It is remarkable that corticosteroid therapy was effective in all treated patients experiencing DIAIH by various drugs (Table 3) and varenicline [136], as opposed to iDILI patients with corticosteroid treatment success in only part of the investigated cohort [184]. This discrepancy implies that iDILI treatment responders may have some uncovered immune or autoimmune features easily accessible to corticosteroid therapy, a view in support of the earlier suggestion that most iDILI cases may have an immune background, while a minority does not have that [179,180].

The varenicline case analyses of iDILI [108,140,141] and DIAIH [136] showed that although varenicline is not metabolized by hepatic microsomal CYPs, alternative pathways including XOR can substitute and are responsible for the liver injury, in line with many other drugs that cause liver injury without activation by CYPs [144]. New also is that autoimmunity as generated by varenicline does not require metabolic CYP pathways [136].

In view of the recent varenicline report [136], a second ALT flare in other suspected iDILI cases may now correctly be attributed to an autoimmune flare, requiring reevaluations of iDILI cases in DILI registries and the LiverTox database or collected by a DILI network. There is also uncertainty whether cases of iDILI cases classified so far as chronic iDILI were indeed DIAIH cases with a protracted clinical course or even progressed to acute liver failure (ALF) with lifesaving liver transplantation, due to missed diagnosis that could have prevented mandatory treatment by corticosteroids. In this context and as a reminder, many patients with ALF of indetermined or indeterminable causes were candidates for liver transplantation that might have been preventable if diagnoses had been established during the initial clinical course [185].

New cases of iDILI and DIAIH attributable to varenicline treatment will hardly be available for further analysis, as Pfizer in Canada halted the production of its Chantix^®^ (varenicline) because of a nitrosamine impurity [185]. Two months later, Pfizer recalled all lots of the drug, making is therefore unavailable permanently, but a generic medication became available in various countries. Varenicline is used as a first-line treatment for smoking cessation and classified by the WHO as an essential drug. As a consequence, a critical evaluation is mandatory concerning the ratio of benefits versus risks, as smoking cessation largely prevents deaths.

Historically, a second ALT peak after drug cessation had already been mentioned as a cautionary signal by the original RUCAM of 1993 [28] and reiterated by the updated RUCAM [30]. It has been argued that a second peak during the usually gradual decline of ALT may have nothing to do with the usual iDILI, but rather suggests that a deviation from normal continuous ALT decline is caused by an alternative process not yet evaluated so far [28], in line with an autoimmune flare [94,136]. A second ALT peak commonly leads to a RUCAM score reduction [30].

### 8.2. Infliximab

Infliximab is highly regarded as a therapeutic option in various autoimmune and inflammatory diseases, including rheumatoid arthritis, Crohn’s disease, ulcerative colitis, ankylosing spondylitis, and psoriatic arthritis [186,187,188,189]. The ratio of benefits over the risks remains positive, although the use of infliximab may be associated with adverse drug reactions [190,191], including liver injury [41,94,138,192,193]. Infliximab shares with varenicline the potential of causing iDILI and DIAIH with a single flare or two flares [94]. The mechanistic steps involved in the development of the two flares occurring after intravenous application of infliximab during the evolution of DIAIH are difficult to elucidate. As opposed to varenicline with its defined chemical structure (Figure 1), infliximab has a much more complex structure [136,194].

Infliximab inhibits by targeting tumor necrosis factor alpha (TNF-α), an essential proinflammatory cytokine involved in the pathogenesis of autoimmune diseases [186]. As a monoclonal antibody and thereby as a protein [192], infliximab is not metabolized by hepatic CYPs or other oxidoreductases, but most probably by unspecific proteases [153]. The metabolic disposition of infliximab is slow, and there is no evidence that infliximab itself does undergo otherwise significant metabolic transformation in the liver: the specific breakdown products are unknown and possibly excreted via the reticuloendothelial system, primarily through macrophages and other immune cells [186]. Serum antidrug antibodies to infliximab in patients under infliximab therapy are signs of active immune reactions [195,196,197]. Overall, however, substantial information gaps exist of possible specific infliximab metabolites and how they and infliximab as the parent drug interact with the native end adaptive immune system and eventually trigger the autoimmune events leading to the second flare of the DIAIH following the intravenous injection of infliximab [94].

## 9. Conclusions

DIAIH is a complex disorder consisting of an acute liver injury and an autoimmunity component. It is commonly diagnosed as uniform liver disease at first presentation of the patient at the office of the family physician or in the hospital. Careful analyses of two patients have revealed that DIAIH can present as a continuous disease with two flares, starting with the acute liver injury and progressing to the hepatic autoimmunity injury, with higher ALT values observed in the first flare compared with the second flare. In the patient with DIAIH due to oral use of varenicline, a smoking cessation drug, the interval between the two flares was two months, much longer than in the patient with DIAIH due to intravenous application of infliximab, who had an interval of around one month only. The metabolic disposition for both drugs was similarly slow and may have contributed to the long interval between the two flares. In both cases, the acute liver injury and the autoimmune liver injury were verified using diagnostic causality algorithms. In the two cases under consideration, it is clear that the DIAIH was not due to preexisting AIH, because autoimmune parameters were not detected at the first flare, but only when the second flare emerged. There are open questions concerning reported iDILI cases with a chronic course that might have been due to an overlooked AIH flare with retarded appearance, as AIH features are often undulant and lack intermittent clinical symptoms. The interval between the two flares classifies the acute liver injury as an initial disruptive event and prerequisite process required for the autoimmune reaction. Notably, the observed interval can create critical problems in the context of DIAIH, iDILI, and AIH. As an example, in cases of iDILI, a subsequent AIH can easily be missed if the clinical course is not carefully evaluated for increased LTs, conditions found among DILI registries and DILI networks, which retrospectively assess iDILI long after the patient has already left the hospital. Clinicians caring for patients with increased LTs under drug therapy should clarify whether it is a traditional iDILI or a DIAIH, perhaps with two flares. It is now the responsibility of scientists dealing prospectively with iDILI patients to determine whether other drugs can produce two flares within a setting of DIAIH.

## Figures and Tables

**Figure 1 ijms-26-09574-f001:**
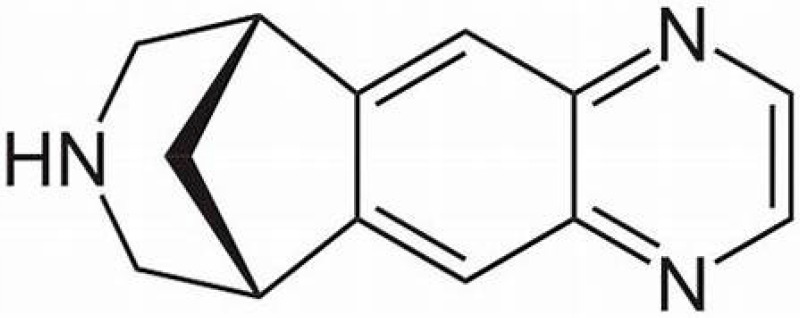
Chemical structure of varenicline.

**Table 1 ijms-26-09574-t001:** Drugs and drug groups implicated in published DIAIH cases with diagnosis verified using the validated causality algorithms of both RUCAM and simplified AIH score.

Drugs and Drug Groups	Cases (*n*)	References
Adalimumab	1 1	Martínez-Casas, 2018 [82] Chung, 2024 [83]
Allopurinol	1	Chung, 2024 [83]
Amitriptyline	1	Weber, 2019 [84]
Amoxicillin-clavulanate	2	García-Cortés, 2023 [85]
Amoxicillin-clavulanate + ceftriaxone	3	Licata, 2014 [86]
Amoxicillin + erythromycin	1	Chung, 2024 [83]
Amoxicillin + metronidazole	1	Chung, 2024 [83]
Anabolic steroid	1	Chung, 2024 [83]
Atorvastatin	2 2 2 1 1	Yeong, 2016 [87] Weber, 2019 [84] García-Cortés, 2023 [85] Tan, 2022 [88] Tse, 2023 [89]
Candesartan	1	Hassoun, 2023 [90]
Cephalexin + amoxicillin	1	Chung, 2024 [83]
Ciprofloxacin	1 1	García-Cortés, 2023 [85] Chung, 2024 [83]
Cyproterone acetate	2	García-Cortés, 2023 [85]
Dabigatran	1	Weber, 2019 [84]
Dexketoprofen	1	García-Cortés, 2023 [85]
Diclofenac	1 2 3	Yeong, 2016 [87] Martínez-Casas, 2018 [82] Weber, 2019 [84]
Ebrotidine	1	García-Cortés, 2023 [85]
Efalizumab	1	García-Cortés, 2023 [85]
Enalapril maleate	1	Hassoun, 2023 [90]
Etanercept	1	Valgeirsson, 2019 [91]
Ezetimibe	1	García-Cortés, 2023 [85]
Fluvastatin	4	García-Cortés, 2023 [85]
Fosfomycin	1	Hassoun, 2023 [90]
Ibandronate	1	Hassoun, 2023 [90]
Ibuprofen	5 1	Hassoun, 2023 [90] García-Cortés, 2023 [85]
Imatinib	1 1 1	Björnsson, 2017 [92] Weber, 2019 [84] Valgeirsson, 2019 [91]
Infliximab	8 7 1 1 1	Björnsson, 2017 [92] Valgeirsson, 2019 [91] Chung, 2024 [83] García-Cortés, 2023 [85] Weber, 2019 [84]
Interferon beta	1	Weber, 2019 [84]
Irbesartan	1	García-Cortés, 2023 [85]
Isotretinoin	1	García-Cortés, 2023 [85]
Lansoprazole	1	Chung, 2024 [83]
Lymecycline	2	Chung, 2024 [83]
Mefenamic acid	1	Hassoun, 2023 [90]
Menotropin	1	Alqrinawi, 2019 [93]
Metamizole	3	Weber, 2019 [84]
Methocarbamol	1	Weber, 2019 [84]
Nimesulide + ketoprofen	6	Licata, 2014 [86]
Minocycline	4 4 1	García-Cortés, 2023 [85] Chung, 2024 [83] Weber, 2019 [84]
NSAIDs + antibiotics	1	Chung, 2024 [83]
Natalizumab	1	Valgeirsson, 2019 [91]
Nitrofurantoin	8 7 5 4 3 1	Martínez-Casas, 2018 [82] Chung, 2024 [83] García-Cortés, 2023 [85] Yeong, 2016 [87] Björnsson, 2017 [92] Hassoun, 2023 [90]
Olmesartan	1	Hassoun, 2023 [90]
Orlistat	1	García-Cortés, 2023 [85]
Pembrolizumab	1	Weber, 2019 [84]
Propylthiouracil	1	Martínez-Casas, 2018 [82]
Rivaroxaban	1	Weber, 2019 [84]
Rosuvastatin	1	García-Cortés, 2023 [85]
Simvastatin	1 1	Yeong, 2016 [87] García-Cortés, 2023 [85]
Sorafenib	1	Tan, 2022 [88]
Trazodone	2	Hassoun, 2023 [90]
Valsartan	1	Hassoun, 2023 [90]

The original RUCAM [28] or its updated version [30] commonly assesses the DILI part, and the AIH part is commonly evaluated by the simplified AIH score [31] or rarely by one of its modifications. Table modified from a previous report published in an open-access journal [2]. Abbreviations: DIAIH, drug-induced autoimmune hepatitis; NSAIDs, nonsteroidal anti-inflammatory drugs; RUCAM, Roussel Uclaf Causality Assessment Method.

**Table 2 ijms-26-09574-t002:** ALT and ALP values and autoimmune parameters described in cases of DIAIH caused by specific drugs and drug groups.

Drugs	Cases (*n*)	ALT (U/L)	ALP (U/L)	Autoimmune Parameters	References
Adalimumab	1	562	NR	ANA	Martínez-Casas, 2018 [82]
Amitriptyline	1	NR	NR	Not specified	Weber, 2019 [84]
Amoxicillin-clavulanate	2	NR	NR	Not specified	García-Cortés, 2023 [85]
Amoxicillin-clavulanate + ceftriaxone	3	NR	NR	Not specified	Licata, 2014 [86]
Amoxicillin + erythromycin	1	NR	NR	Not specified	Chung, 2024 [83]
Amoxicillin + metronidazole	1	NR	NR	Not specified	Chung, 2024 [83]
Anabolic steroid	1	NR	NR	Not specified	Chung, 2024 [83]
Atorvastatin	2 2 2 1 1	721 NR NR 696 385	NR NR NR 107 163	ANA, ASMA Not specified Not specified Unremarkable ANA	Yeong, 2016 [87] Weber, 2019 [84] García-Cortés, 2023 [85] Tan, 2022 [88] Tse, 2023 [89]
Candesartan	1	NR	NR	Not specified	Hassoun, 2023 [90]
Cefalexin + amoxicillin	1	NR	NR	Not specified	Chung, 2024 [83]
Ciprofloxacin	1	NR	NR	Not specified	García-Cortés, 2023 [85]
Cyproterone acetate	2	NR	NR	Not specified	García-Cortés, 2023 [85]
Dabigatran	1	NR	NR	Not specified	Weber, 2019 [84]
Dexketoprofen	1	NR	NR	Not specified	García-Cortés, 2023 [85]
Diclofenac	1 2 3	3489 1491 NR	NR NR NR	ANA, ASMA ANA, ASMA, SLA Not specified	Yeong, 2016 [87] Martínez-Casas, 2018 [82] Weber, 2019 [84]
Ebrotidine	1	NR	NR	Not specified	García-Cortés, 2023 [85]
Efalizumab	1	NR	NR	Not specified	García-Cortés, 2023 [85]
Ezetimibe	1	NR	NR	Not specified	García-Cortés, 2023 [85]
Enalapril maleate	1	NR	NR	Not specified	Hassoun, 2023 [90]
Fluvastatin	4	NR	NR	Not specified	García-Cortés, 2023 [85]
Fosfomycin	1	NR	NR	Not specified	Hassoun, 2023 [90]
Ibandronate	1	NR	NR	Not specified	Hassoun, 2023 [90]
Ibuprofen	5 1	NR NR	NR NR	Not specified Not specified	Hassoun, 2023 [90] García-Cortés, 2023 [85]
Imatinib	1 1	1212 NR	205 NR	ANA Not specified	Björnsson, 2017 [92] Weber, 2019 [84]
Infliximab	10 8 1 1	1658 NR NR NR	493 NR NR NR	ANA ASMA Not specified Not specified	Björnsson, 2017 [92] Valgeirsson, 2019 [91] García-Cortés, 2023 [85] Weber, 2019 [84]
Interferon beta	1	NR	NR	Not specified	Weber, 2019 [84]
Irbesartan	1	NR	NR	Not specified	García-Cortés, 2023 [85]
Isotretinoin	1	NR	NR	Not specified	García-Cortés, 2023 [85]
Lansoprazole	1	NR	NR	Not specified	Chung, 2014 [83]
Lymecycline	2	NR	NR	Not specified	Chung, 2024 [83]
Mefenamic acid	1	NR	NR	Not specified	Hassoun, 2023 [90]
Menotropin	1	504	366	ANA	Alqrinawi, 2019 [93]
Metamizole	3	NR	NR	Not specified	Weber, 2019 [84]
Methocarbamol	1	NR	NR	Not specified	Weber, 2019 [84]
Minocycline	4 4 1	NR NR NR	NR NR NR	Not specified Not specified Not specified	García-Cortés, 2023 [85] Chung, 2024 [83] Weber, 2019 [84]
Nimesulide + ketoprofen	6	NR	NR	Not specified	Licata, 2014 [86]
Nitrofurantoin	8 7 5 4 3 1	2059 NR NR 587 1974 NR	NR NR NR NR 204 NR	ANA, ASMA Not specified Not specified ANA, ASMA ANA Not specified	Martínez-Casas, 2018 [82] Chung, 2024 [83] García-Cortés, 2023 [85] Yeong, 2016 [87] Björnsson, 2017 [92] Hassoun, 2023 [90]
NSAIDs + antibiotics	1	NR	NR	Not specified	Chung, 2024 [83]
Olmesartan	1	NR	NR	Not specified	Hassoun, 2023 [90]
Orlistat	1	NR	NR	Not specified	García-Cortés, 2023 [85]
Pembrolizumab	1	NR	NR	Not specified	Weber, 2019 [84]
Propylthiouracil	1	754	NR	ANA	Martínez-Casas, 2018 [82]
Rivaroxaban	1	NR	NR	Not specified	Weber, 2019 [84]
Rosuvastatin	1	NR	NR	Not specified	García-Cortés, 2023 [85]
Simvastatin	1 1	1245 NR	NR NR	ANA, ASMA Not specified	Yeong, 2016 [87] García-Cortés, 2023 [85]
Sorafenib	1	1004	190	Unremarkable	Tan, 2022 [88]
Trazodone	2	NR	NR	Not specified	Hassoun, 2023 [90]
Valsartan	1	NR	NR	Not specified	Hassoun, 2023 [90]

The original RUCAM [28] or its updated version [30] commonly assessed the DILI part, and the AIH part was commonly evaluated by the simplified AIH score [31] or rarely by one of its modifications. List derived from a previous report published in an open-access journal [2]. Abbreviations: ANA, anti-nuclear antibody; ASMA, anti-smooth muscle antibody; DIAIH, drug-induced autoimmune hepatitis; NR, not reported; NSAIDs, anti-inflammatory drugs; RUCAM, Roussel Uclaf Causality Assessment Method; SLA, soluble liver antigen antibody.

**Table 3 ijms-26-09574-t003:** Selected drugs implicated in DIAIH with response after cessation of the suspected drug following therapy with unspecified immunosuppressants or after specific treatment with prednisolone and azathioprine followed by tacrolimus and ursodeoxycholic acid.

Drugs	Cases (*n*)	Response to Drug Cessation/Therapy	References
Adalimumab	1 1	CR with PRED/AZA CR with cessation of the culprit drug	Martínez-Casas, 2018 [82] Chung, 2024 [83]
Allopurinol	1	CR with IS	Chung, 2024 [83]
Amoxicillin + erythromycin	1	CR with IS	Chung, 2024 [83]
Amoxicillin + metronidazole	1	CR with IS	Chung, 2024 [83]
Anabolic steroid	1	CR with IS	Chung, 2024 [83]
Atorvastatin	1 2	CR with PRED CR with cessation of the culprit drug	Tan, 2022 [88] Tse, 2023 [89]
Cefalexin + amoxicillin	1	CR with IS	Chung, 2024 [83]
Ciprofloxacin	1	CR with IS	Chung, 2024 [83]
Diclofenac Diclofenac + ibuprofen	1 1	CR with PRED/AZA IR with PRED/AZA/TAC/UCDA	Martínez-Casas, 2018 [82] Chung, 2024 [83]
Infliximab	1 2	CR with IS CR with cessation of the culprit drug	Chung, 2024 [83] Chung, 2024 [83]
Lansoprazole	1	CR with IS	Chung, 2024 [83]
Menotropin	1	CR with PRED/AZA	Alqrinawi, 2019 [93]
Minocycline	1	CR with IS	Chung, 2024 [83]
NSAIDs + Antibiotics	1	CR with IS	Chung, 2024 [83]
Nitrofurantoin	8 7	CR with PRED/AZA CR with IS	Martínez-Casas, 2018 [82] Chung, 2024 [83]
Propylthiouracil	1	CR with PRED/AZA	Martínez-Casas, 2018 [82]
Sorafenib	1	CR with cessation of the culprit drug	Tan, 2022 [88]

Compilation of selected drugs implicated in causing DIAIH with specification of therapy modalities and their efficacies. The DILI part of DIAIH was commonly assessed by the original RUCAM [28] or its updated version [30], while the AIH part was commonly evaluated by the simplified AIH score [31] or rarely by one of its modifications. The table was modified from a previous report published in an open-access journal [2]. Abbreviations: AZA, azathioprine; CR, complete response; DIAIH, drug-induced autoimmune hepatitis; IR, incomplete response; IS, immunosuppressant, not further specified; NSAIDs, nonsteroidal anti-inflammatory drugs; PRED, prednisolone; RUCAM, Roussel Uclaf Causality Assessment Method; TAC, tacrolimus; UCDA, ursodeoxycholic acid.

**Table 4 ijms-26-09574-t004:** Selected drugs implicated in suspected DIAIH with unverified diagnosis.

Drugs	Cases (*n*)	RUCAM Causality Algorithm Used	Simplified Criteria of AIH Score Used	DIAIH Diagnosis Verified by Both RUCAM and Simplified AIH Score	Literature
Adalimumab	1 1	Yes No	No Yes	No No	Ghabril, 2013 [94] Rodrigues, 2015 [95]
Atorvastatin	1	Yes	No	No	Khan, 2020 [96]
Cephalexin	1	No	Yes	No	Björnsson, 2010 [97]
Etanercept	2	Yes	No	No	Ghabril, 2013 [94]
Hydralazine	7	No	No	No	de Boer, 2017 [98]
Infliximab	25 8 3	Yes No Yes	No Yes No	No No No	Björnsson, 2022 [99] Rodrigues, 2015 [95] Ghabril, 2013 [94]
Methyldopa	10	No	No	No	de Boer, 2017 [98]
Minocycline	19 10 1	No No No	No Yes No	No No No	de Boer, 2017 [98] Björnsson, 2010 [97] Harmon, 2018 [100]
Nitrofurantoin	24 10	No No	No Yes	No No	de Boer, 2017 [98] Björnsson, 2010 [97]
Pirfenidone	1	Yes	No	No	Fortunati, 2024 [101]
Prometrium	1	No	Yes	No	Björnsson, 2010 [97]

Compilation of suspected drugs implicated in causing DIAIH, but without complete diagnostic causality verification by valid methods. For some patients, the DILI part of DIAIH was causally evaluated solely by the original RUCAM [28] or the updated RUCAM [30], while other cases had the AIH part assessed solely by the simplified AIH score [31] or rarely by one of its modifications. A group of DIAIH patients were evaluated even by none of the methods. Table modified and derived from a previous report published in an open-access journal [2]. Abbreviations: DIAIH, drug-induced autoimmune hepatitis; NSAIDs, nonsteroidal anti-inflammatory drugs; RUCAM, Roussel Uclaf Causality Assessment Method.

## Data Availability

No new data were created or analyzed in this study. Data sharing is not applicable to this article.

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
