# Peer review of "Drug-Induced Autoimmune Hepatitis by Varenicline and Infliximab as a Continuous Disease Spectrum with Two Different Flares: Acute Liver Injury Followed by Hepatic Autoimmunity"

_ijms, 2025, doi:10.3390/ijms26199574_

Round 1

Reviewer 1 Report

Comments and Suggestions for Authors

This review focuses on drug-induced autoimmune hepatitis (diaih), a rare and complex disease, focusing on the analysis of its rare "double attack" clinical features, and combing the types of drugs that may cause diaih, treatment and prognosis.The following points still need attention:
1. it is suggested to standardize the use of abbreviations and full names of key terms in the full text, "anti smooth muscle antibodies" is sometimes abbreviated as "ASMA" and sometimes expressed as "SMA"; "Simplified autoimmune responsibilities score" is sometimes omitted in different paragraphs and is directly called "AIH score".
2. the logic of some paragraphs is divergent. For example, the analysis defect of livertox database and the production stop information of vancomland are inserted halfway, resulting in the core issues are not prominent enough. The logic and expression of paragraphs can be further modified.
3. the article focuses on the case analysis and mechanism research of diaih, but the practical guiding significance for clinicians is slightly insufficient. This content can be added to the discussion part.

Author Response

Dear Reviewer 1,

Thank you for valuable suggestions, virtually of these are considered in the revision.

This review focuses on drug-induced autoimmune hepatitis (diaih), a rare and complex disease, focusing on the analysis of its rare "double attack" clinical features, and combing the types of drugs that may cause diaih, treatment and prognosis.The following points still need attention:
1. it is suggested to standardize the use of abbreviations and full names of key terms in the full text, "anti smooth muscle antibodies" is sometimes abbreviated as "ASMA" and sometimes expressed as "SMA"; "Simplified autoimmune responsibilities score" is sometimes omitted in different paragraphs and is directly called "AIH score".                                                                     Thank you, standardization was done.
2. the logic of some paragraphs is divergent. For example, the analysis defect of livertox database and the production stop information of vancomland are inserted halfway, resulting in the core issues are not prominent enough. The logic and expression of paragraphs can be further modified.                                                                                                                           Half way is prominent enough, a new para or expanded discussion  is not warranted.  
3. the article focuses on the case analysis and mechanism research of diaih, but the practical guiding significance for clinicians is slightly insufficient. This content can be added to the discussion part.                                                                                                                                        Addition was done at end of conclusions.

Reviewer 2 Report

Comments and Suggestions for Authors

This is an extensive review from Teschke R, on the DILI from varenicline and infliximab analyzing the classifying scores for the diagnosis of DILI and the pathophysiological mechanisms for DILI mediated from these drugs.

This an extensive manuscript with a lot of information. However, I have some serious considerations about the manuscript. First of all, the manuscript is very long and difficult to follow. Second it uses the terminology of DIAIH, while this is substituted by the term drug induced autoimmune like hepatitis (DIALH) to describe cases of DILI with autoimmune features. Moreover, it refers to classifying tools as RUCAM and simplified. Simplified score for AIH does not include the probability of DILI, although revised score yes, while the use of RECAM which is a revised version of RUCAM is not mentionned. 

Afterwards, it is no clear why the author has decided to focus on DILI mediated from vareniline and infliximab, and I cannot help but wonder why the study referring to the relapse of DILI from vareniline is considered as of high importance while there are a lot of studies with same results as well as international guidelines that emphasize on the fact that a relapse of DIALH should be considered and treated as AIH.

Finally I think the tone is kind of imperative in some parts an I would suggest to be a more gentle.

Overall, I think that the manuscript includes a lot and valuable information but the aims and the design should be clear and improved respectively.

Author Response

Dear Reviewer 2,

Thank you for your constructive comments, virtually all of these are considered in the revision.

This is an extensive review from Teschke R, on the DILI from varenicline and infliximab analyzing the classifying scores for the diagnosis of DILI and the pathophysiological mechanisms for DILI mediated from these drugs.

This an extensive manuscript with a lot of information. However, I have some serious considerations about the manuscript. First of all, the manuscript is very long and difficult to follow. Second it uses the terminology of DIAIH, while this is substituted by the term drug induced autoimmune like hepatitis (DIALH) to describe cases of DILI with autoimmune features. Moreover, it refers to classifying tools as RUCAM and simplified. Simplified score for AIH does not include the probability of DILI, although revised score yes, while the use of RECAM which is a revised version of RUCAM is not ntionned. The term DIAIH is correct and used now in many reports. Since the simplified AIH score does not consider DILI, the updated RUCAM is required. RECAM was not used in any of the referenced papers. So, no need to discuss this tool lacking validation.     

Afterwards, it is no clear why the author has decided to focus on DILI mediated from vareniline and infliximab, and I cannot help but wonder why the study referring to the relapse of DILI from vareniline is considered as of high importance while there are a lot of studies with same results as well as international guidelines that emphasize on the fact that a relapse of DIALH should be considered and treated as AIH.                                                                                                         DIAIH by varenicline and infliximab are excellent examples that the relapse occurred not due to a pre-existing AIH. At the first flare, no autoimmune parameters were detected, commented now in the middle of the Conclusion section.    

Finally I think the tone is kind of imperative in some parts an I would suggest to be a more gentle. Ok I understand your concern and, to avoid delay of publication, I toned down: L268,269,322,323,328,329,333-335.                                                                              

Nevertheless, clear words are needed in scientific papers, to be addressed especially to those who were careless regarding case data evaluation and were harsh, unjustified,  and less gentle to non-US scientists, but that’s part of the scientific game. Clear words will hopefully help avoid basic errors and misconceptions in future cases.

Overall, I think that the manuscript includes a lot and valuable information but the aims and the design should be clear and improved respectively. Done in abstract P1 lines 13-16.